# A nitrogen-doped nanotube molecule with atom vacancy defects

Koki Ikemoto [1,2], Seungmin Yang [1], Hisashi Naito [3], Motoko Kotani[4,5], Sota Sato [1,2] & Hiroyuki Isobe [1,2✉]

Nitrogen-doped carbon nanotubes have attracted attention in various fields, but lack of congeners with discrete molecular structures has hampered developments based on in-depth, chemical understandings. In this study, a nanotube molecule doped periodically with multiple nitrogen atoms has been synthesized by combining eight 2,4,6-trisubstituted pyridine units with thirty-two 1,3,5-trisubstituted benzene units. A synthetic strategy involving geodesic phenine frameworks is sufficiently versatile to tolerate pyridine units without requiring synthetic detours. Crystallographic analyses adopting aspherical multipole atom models reveal the presence of axially rotated structures as a minor disordered structure, which also provides detailed molecular and electronic structures. The nitrogen atoms on the nanotube serve as chemically distinct sites covered with negatively charged surfaces, and they increase the chance of electron injections by lowering the energy levels of the unoccupied orbitals that should serve as electron acceptors.

[1] Department of Chemistry, The University of Tokyo, Hongo, Tokyo 113-0033, Japan. [2] JST, ERATO, Isobe Degenerate $\pi$-Integration Project, Hongo, Tokyo 113-0033, Japan. [3] Graduate School of Mathematics, Nagoya University, Nagoya 464-8602, Japan. [4] Advanced Institute for Materials Research, Tohoku University, Sendai 980-8577, Japan. [5] Mathematical Institute, Tohoku University, Sendai 980-8578, Japan. ✉email: isobe@chem.s.u-tokyo.ac.jp

The uniqueness of nanoscale graphitic networks is being exploited, and, in particular, nanocarbons containing non-carbon elements attract attention. Nitrogen-doped carbon nanotubes were the first examples of doped nanocarbons[1–3], and modulations of electronic structures were demonstrated[4–6]. Although donor-type contributions of nitrogen dopants have been taken for granted, due to preceding concepts of doped n-type inorganic semiconductors[7], effects of nitrogen-doped nanotubes are still controversial[4–6] because of the lack of sufficient information about atomic-level and electronic structures[8]. Here we show the bottom-up synthesis and high-precision structures of a nitrogen-doped nanotube molecule with a discrete, rigid structure. Taking advantage of a versatile synthetic strategy[9,10], we replace eight methine (CH) groups of a (12,12)-phenine nanotube (pNT) molecule[11] with eight nitrogen atoms and eight atom vacancy sites (Fig. 1). Detailed molecular and electronic structures are disclosed to reveal unforeseen effects of nitrogen doping.

## Results

**Synthesis**. By replacing the phenine unit of pNT with 2,4,6-tri-substituted pyridine, i.e., a nitrogen-doped phenine congener, we designed nitrogen-doped pNT molecule (NpNT)[11]. A few minor modifications were made in the synthesis, which resulted in a slightly improved overall yield for the nitrogen-doped variant (Fig. 2). Thus, the starting material was changed from dibromobenzene to 1-bromo-3-chlorobenzene, which allowed the coupling reaction with 2,6-dibromopyridine (**4**) via silylation and borylation, and a resultant terphenylene congener (**5**) was coupled by a Yamamoto-type coupling reaction to afford a nitrogen-doped congener of [6]cyclo-*meta*-phenylene ([6]CMP) (**6**) with an improved yield. The nitrogen-doped [6]CMP congener was then cyclized to a flexible cyclic GPF (**8**) through borylation, Pt-mediated macrocyclization, ligand exchange and reductive elimination[12]. The final three steps of iododesilylation, Suzuki–Miyaura coupling and Yamamoto-type coupling completed the nanometre-sized cylinder molecule, (12,12)-NpNT. A minor modification of the catalyst for Suzuki–Miyaura coupling was also made, and the overall yield of (12,12)-NpNT was improved to 1.4% for the 10-step synthesis from the value of 0.7% that was recorded for the 9-step synthesis of (12,12)-pNT[11]. The structure of (12,12)-NpNT was first established by spectroscopy. The $^1$H NMR spectrum showed seven aromatic singlet resonances (Fig. 2), which showed the loss of one methine resonance

relative to the eight aromatic resonances of (12,12)-pNT[11]. The chemical composition of $C_{296}H_{256}N_8$ was confirmed by a MALDI-TOF mass spectrum showing the presence of an ionized species with $m/z = 3923$ [M + H]$^+$. In the chemical composition, the nitrogen atoms occupy 2.6% of non-hydrogen atoms (8/304), which is close to the nitrogen contents often reported for nitrogen-doped carbon nanotubes (2–5%)[2,4]. Representative geometric descriptions of the molecule are summarized as follows. The (12,12)-NpNT molecule possesses a graphitic lattice of (12,12)-carbon nanotube with a length index of $t_f = 7.0$ (ref. [13]). The structural defects comprise both replacements and depletions of atoms and bonds (Supplementary Fig. 1), which can be quantitatively described by geometric measures of bond-filling and atom-filling indices of $F_b = 53\%$ and $F_a = 64\%$. By using an oblique coordinate system of carbon nanotubes (see the Supplementary Methods for details)[13,14], we can further identify and describe the positions of nitrogen atoms as (5,–2), (8,1), (11/3, –10/3), (11,4), (14,7), (20/3,–1/3), (29/3,8/3), and (38/3,17/3) (Supplementary Fig. 2).

**Crystallography**. For the elucidation of precise molecular structures, elaborate X-ray diffraction analyses were necessary because of unexpected, disordered structures unique to the cylindrical shape. A single crystal of (12,12)-NpNT was obtained from a 1,2-dichloroethane solution by diffusing 2-propanol vapour at 25 °C. With 22,961 unique reflections from 299,140 total reflections (multiplicity of observations = 13) from a single crystal, we first solved the structure by a conventional analytical method adopting spherical, independent atom models (IAM)[15]. With a few *t*-Bu conformations located as disordered structures, the structure converged with a moderate R-factor [$R(F^2) = 0.1267$ on SHELX[16] and $R(F) = 0.1010$ on XD2016][17] (Supplementary Fig. 4, stage 0 and stage 1). To define a structure with a higher accuracy, we adopted multipole aspherical atom models in the Hansen–Coppens formalism[18] by using a transferrable aspherical atom model (TAAM)[19] with parameters from the University at Buffalo pseudoatom databank (UBDB)[20] (Supplementary Fig. 3). Close examinations of the residual electron densities ($F_o$–$F_c$) after the TAAM analysis indicated the presence of a disordered structure that originated from a 45° axial rotation of the nanometre-sized cylinder (Supplementary Fig. 4; stage 2). Such rotational disorders were not found with a previous example of hydrocarbon pNT[11]. With this anomalous rotational disorder included as a minor structure (occupancy = 12%), the molecular structures were solved with a higher accuracy (Fig. 3a): the $R(F)$ values reached 0.0855 with IAM/XD2016 analysis (Supplementary Fig. 4, stage 3), which was further improved to $R(F) = 0.0802$ with TAAM/XD2016 analysis with multipole aspherical atom models (Supplementary Fig. 4, stage 4). The $F_o$–$F_c$ maps confirmed the decrease in the residual electron densities (Supplementary Fig. 4), and the deformation maps of $F_1$–$F_2$ confirmed the deformed densities of the major structure. The two axial rotational disorders were made possible by the cylindrical shape with uniform diameters (1.6 nm; Supplementary Fig. 5). The nanometre-sized cylindrical structure is shown for the major disorder (occupancy = 88%) in Fig. 3b. The introduction of eight nitrogen atoms did not affect the overall molecular structures, and negligible differences between (12,12)-NpNT and (12,12)-pNT were noted with the π-orbital axis vector (POAV) $\theta_p$ and dihedral angles[21] (Supplementary Fig. 6).

**Electronic structures**. The effects of the nitrogen dopants were determined from the fine molecular structures. As observed from the crystal structure of 4,4′-bipyridine, the lengths of C–N bonds in the aromatic ring are shorter than the lengths of the C–C

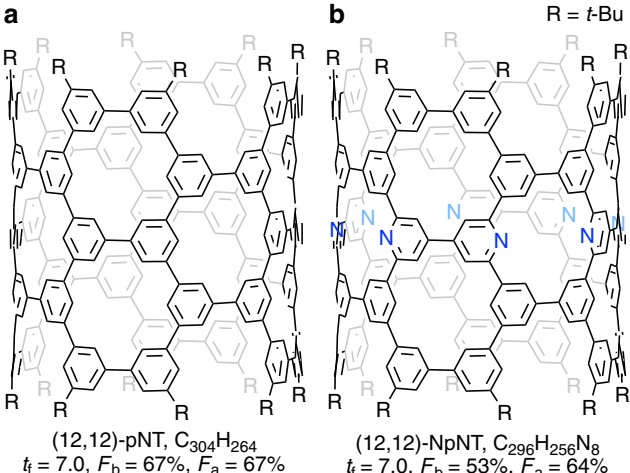

**a** (12,12)-pNT, $C_{304}H_{264}$
$t_f = 7.0$, $F_b = 67\%$, $F_a = 67\%$

**b** (12,12)-NpNT, $C_{296}H_{256}N_8$
$t_f = 7.0$, $F_b = 53\%$, $F_a = 64\%$

R = *t*-Bu

**Fig. 1 Phenine nanotube molecules. a** (12,12)-pNT. **b** Nitrogen-doped (12,12)-NpNT.

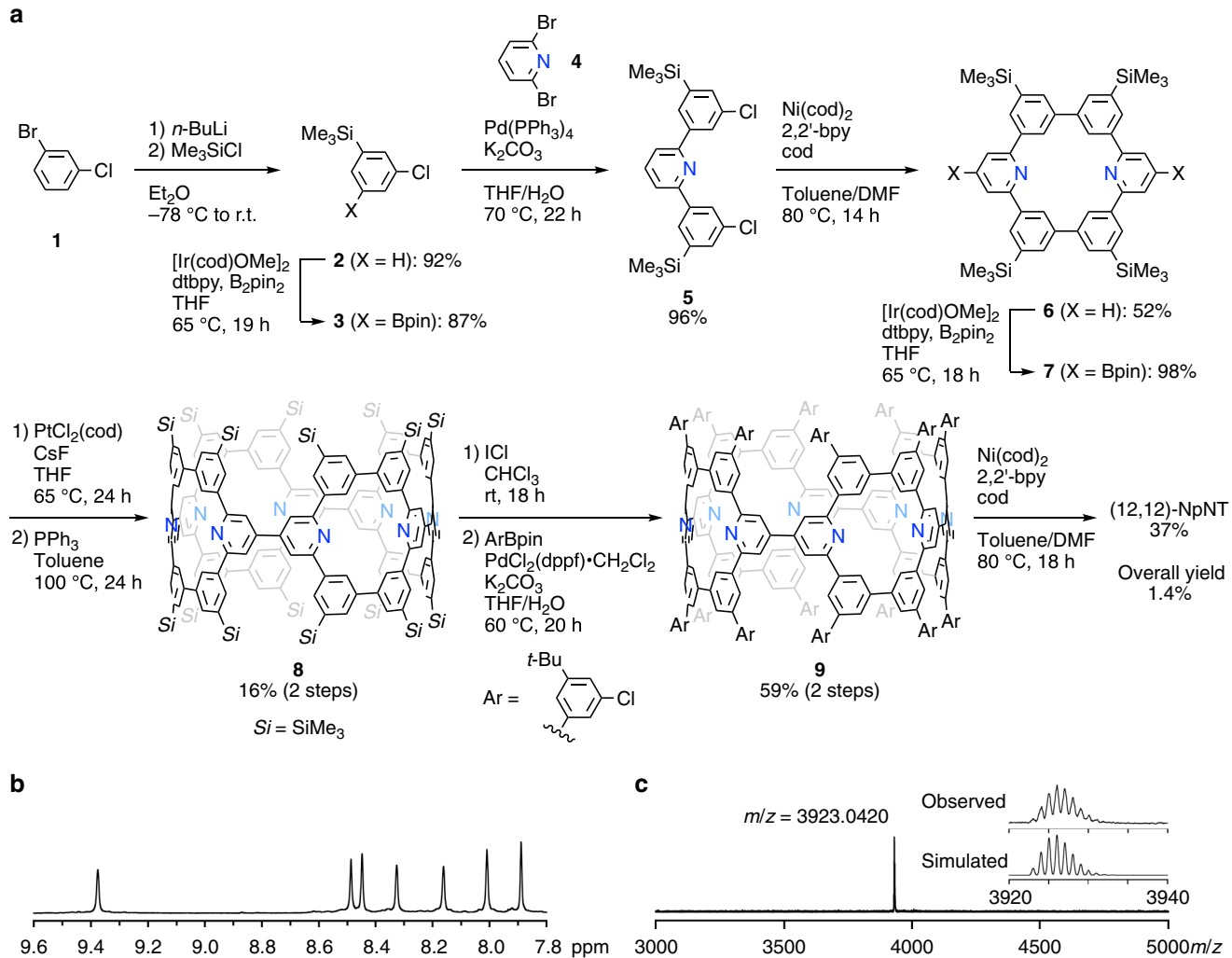

**Fig. 2 Synthesis of (12,12)-NpNT. a** Synthesis. **b** A proton NMR spectrum in CDCl₃ at 298 K. **c** A MALDI-TOF mass spectrum (dithranol matrix, positive).

bonds (1.34 vs. 1.39 Å) (Supplementary Fig. 7). Unlike flexible cycloarylenes with fluctuating structures[22,23], (12,12)-NpNT possessed a rigid molecular structure, which allowed for unequivocal assignments of the nitrogen atoms. The crystallographic analyses of (12,12)-NpNT thus allocated the shorter bonds of 1.33 Å at the expected positions to connect the nitrogen atoms (Supplementary Fig. 7). The effects of nitrogen atoms and their bonding on the electron density were further clarified by charge-density analysis after TAAM/XD2016 refinements[18,19]. The deformation map showing the electron density deformed by chemical bonds thus revealed the lone pair densities on the nitrogen atoms (Fig. 3c). An electrostatic potential map also revealed the charge distributions and located the red areas of higher electron densities around the nitrogen atoms, characterizing the nitrogen sites as chemically distinct areas with an abundance of electrons (Fig. 3d).

**Optical properties**. The effects of nitrogen doping were also observed from the optical properties. As shown in Fig. 4a, the UV absorption spectrum of (12,12)-NpNT was redshifted from that of non-doped (12,12)-pNT ($\lambda_{edge}$ = 345 nm)[11], with the edge absorption appearing at 378 nm (optical gap = 3.28 eV). Fluorescence was also observed with (12,12)-NpNT with a quantum yield of 16%. The density functional theory (DFT) calculations of (12,12)-NpNT also confirmed the narrowed gap between the highest occupied molecular orbital (HOMO) and the lowest

unoccupied molecular orbital (LUMO) (3.07 eV) with highly degenerate orbitals (Fig. 4b). However, the lone pairs of nitrogen did not contribute to the narrowing of the gap, with HOMO-7 to HOMO-10 (−3.55 eV) located below HOMO to HOMO-6 among the conjugated π-orbitals (−3.48 eV) (Fig. 4b and Supplementary Fig. 9). In contrast, the nitrogen-related π*-orbitals were inserted on the unoccupied orbital side as LUMO to LUMO+3 at −0.41 to −0.38 eV, which narrowed the HOMO–LUMO gap of the cylindrical molecule. The effects of nitrogen atoms over the electronic structures of NpNT were unexpected from HOMO/LUMO-stabilizations found with N-doped acenes[24].

The optical properties were modulated under acidic conditions. When we added an excess amount of trifluoroacetic acid, the absorption and fluorescence spectra were broadened (Fig. 4d). Moreover, when photoluminescence quantum yields were measured, the quantum yield dropped from 16% to 8% under the acidic conditions. We believe that the spectral changes can be ascribed, most likely, to the protonation at the pyridinic nitrogen atoms of NpNT under acidic conditions.

**Discussion**

A nitrogen-doped congener of phenine nanotubes was synthesized, which also allowed us to periodically embed atom vacancy defects into a nanotube molecule. Crystallographic analyses adopting aspherical atomic models revealed the presence of unique disorders with axially rotated molecular structures, which

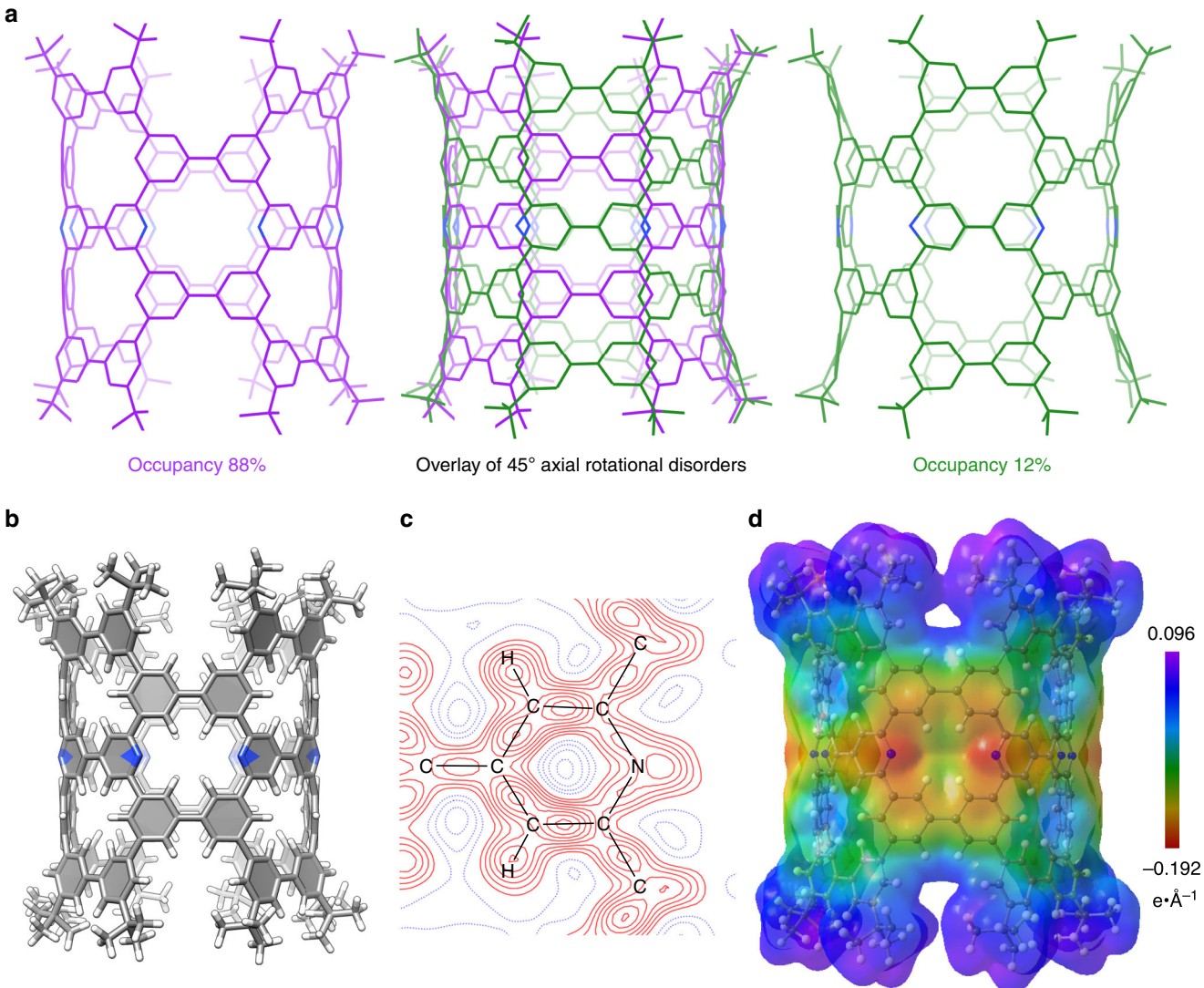

**Fig. 3 Crystal structures of (12,12)-NpNT. a** Molecular structures of two disordered structures. **b** Molecular structure (occupancy = 88%). **c** Deformation map (contour interval: 0.02 e Å$^{-3}$, positive: red, negative: blue). Note that, because of the lower occupancy of 12%, negligible densities of the minor disordered structure were found in the electron and deformation density map, which also resulted in distortions of the assigned structures. See also Fig. 3a and Supplementary Fig. 5. **d** Electrostatic potential map mapped on the 0.0067 e Å$^{-3}$ isosurface of the electron density.

should similarly exist in bundles of carbon nanotubes. The aspherical atom models also allowed charge density analyses, showing the subtle yet apparent effects of nitrogen atoms on the structures. Interestingly, the nitrogen lone pairs created chemically distinct, basic sites on the molecule but did not affect the HOMO level. In contrast, the nitrogen-related π*-orbitals lowered the LUMO level, which should be favourable in n-type organic semiconductors by accepting negative charge carriers. Thus, unlike the inorganic semiconductors where lone pairs donate charge carriers in the conduction bands, the nitrogen-doped nanotubes tend to be n-type organic semiconductors where lowered unoccupied orbitals facilitate electron injections from the electrodes. Thus, this study revealed unique characteristics of so-called pyridinic nitrogen atoms doped on graphitic networks[4–6]. This study also demonstrated that the structure–property relationships of nanocarbon molecules, particularly those with rigid discrete structures, can deepen our understanding of the effects of heteroatom doping[25]. Taking advantage of the versatile synthetic approach of phenine frameworks, we will explore other variants of heteroatom dopants with various locations and structures of dopants in the near future.

## Methods

**Synthesis**. The nitrogen-doped (12,12)-NpNT was synthesized by the procedure detailed in the Supplementary Methods. All the data necessary for the identification are also provided.

**Location of defects**. Locations of defects were defined by using an oblique coordinate system of carbon nanotubes[14]. Details such as the location of the origin are described in the Supplementary Methods. The defect locations can also be defined by using a web-based applet at https://physorg.chem.s.u-tokyo.ac.jp/applet/defect/.

**Crystallography**. A single crystal (ca. 0.04 × 0.03 × 0.02 mm$^3$) suitable for the crystallographic analysis was obtained by slowly diffusing 2-propanol vapour into a solution of (12,12)-NpNT in 1,2-dichloroethane at 25 °C. A single crystal was mounted on a thin polymer tip with cryoprotectant oil and frozen at –173 °C via flash-cooling. The diffraction analysis with a synchrotron X-ray source was conducted at –173 °C at the beamline BL26B1 (SPring-8), which was collimated for an increased brightness with a capillary lens (Hamamatsu, J12432-01). The beamline for macromolecular crystallography was equipped with a single-axis goniometer (KOHZU, QKSU-SDD) with an additional house-build compact κ goniometer head for the crystal tilt. Diffraction data were collected by a hybrid photon counting detector (Dectris EIGER X 4M) comprising 2070 × 2167 pixels up to sin θ λ$^{-1}$ = 0.60 Å$^{-1}$ resolution[26]. The first set of diffractions were collected with an oscillation range of 0.1° (exposure time = 0.3 s) for a ω range of 360°. Because of an insufficient level of the completeness (94.6%), a second set of 360°-diffractions were collected after tilting the crystal by 30°. Two datasets were merged with the XDS

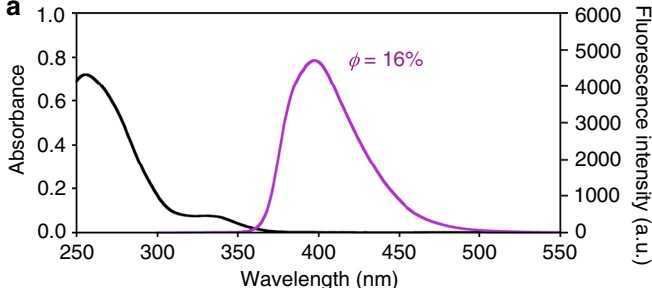

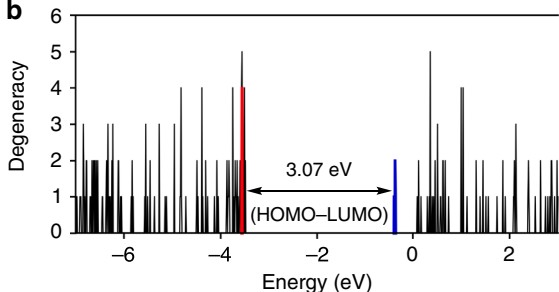

3.07 eV

(HOMO–LUMO)

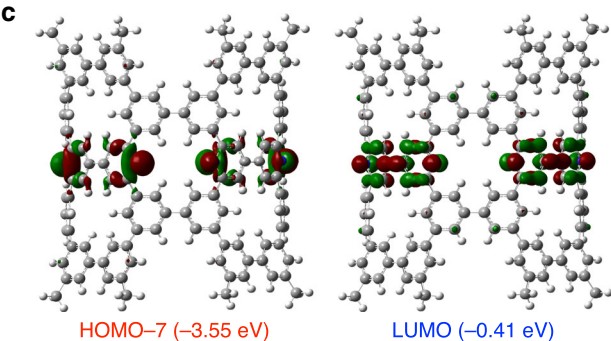

HOMO–7 (–3.55 eV)     LUMO (–0.41 eV)

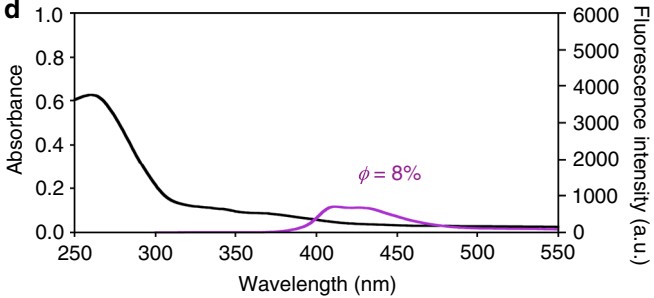

**Fig. 4 Representative properties of (12,12)-NpNT. a** Absorption and fluorescence spectra of (12,12)-NpNT ($1.2 \times 10^{-6}$ M in $CH_2Cl_2$, 25 °C). **b** Histogram sorting 213 orbitals of (12,12)-NpNT by energy on the basis of the DFT calculations. Orbitals originating from pyridine units are coloured in red (occupied) and blue (unoccupied). See Supplementary Fig. 9 for details. **c** Representative Kohn–Sham orbitals. **d** Absorption and fluorescence spectra of (12,12)-NpNT ($1.2 \times 10^{-6}$ M in $CH_2Cl_2$, 25 °C) in the presence of trifluoroacetic acid ($7.4 \times 10^{-3}$ M).

programme to afford 299,140 total reflections with 22,961 unique reflections (multiplicity of observations = 13)[27]. After solving the initial structure by a direct method by using SHELXT software programme[28], the final structure was obtained through trial-and-error processes adopting IAM/SHELXL[16], IAM/XD2016 and TAAM/XD2016 refinement protocols[17]. In short, the first candidate structure with *t*-Bu disorders was obtained by IAM/SHELXL and IAM/XD2016 refinements but was partly refuted by TAAM/XD2016 refinements with inferior R values. The residual electron densities after the IAM/XD2016 refinements indicated the presence of an unexpected disordered cylinder that was located at axially rotated

orientations. By taking account of the minor rotated disorder, we then finalized the structures through IAM/SHELXL, IAM/XD2016 and TAAM/XD2016 refinements. Further details of the refinements and analyses are described in the Supplementary Methods. In addition to our final cif file (CCDC 1966650), we also deposited a preliminary cif file at the stage 0 as CCDC 1984802. The res and HKL data embedded in this file can allow readers to follow the present analytical procedures from the raw data.

**DFT calculations**. The DFT calculations were performed by the same method used for (12,12)-pNT[11]. The geometry optimizations were thus performed at the PBEPBE/STO-3G level of theory[29,30] by using Gaussian 16 programme suite[31].

**Optical properties**. A solution of (12,12)-NpNT was prepared in $CH_2Cl_2$ at $1.2 \times 10^{-6}$ M. The absorption and fluorescent spectra were recorded at 25 °C on V-670 (JASCO) and FP-8500 (JASCO; excitation = 255 nm) spectrometers, and the photoluminescence quantum yield was determined by C9920-02G spectrometer (Hamamatsu; excitation = 255 nm). Fluorescence lifetime was determined as 3.3 ns on Quantaurus-QY C11347 (Hamamatsu). The same spectra were also recorded after the addition of trifluoroacetic acid ($7.4 \times 10^{-3}$ M).

## Data availability

Synthetic and experimental procedures, as well as crystallographic, spectroscopic and computational data are provided in the Supplementary Information. Crystallographic data for the structures reported in this Article have been deposited at the Cambridge Crystallographic Data Centre, under deposition numbers CCDC 1966650, CCDC 1966651 and CCDC 1984802. Copies of the data can be obtained free of charge via www. ccdc.cam.ac.uk/data_request/cif. For the designation of defect locations, we provide a web-based applet that can be used free of charge at https://physorg.chem.s.u-tokyo.ac.jp/applet/defect/.

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

## Acknowledgements

We thank Professors B. Dittrich (Heinrich-Heine-Universität Düsseldorf) and T. Koritsanszky (Middle Tennessee State University) for their kind advices on TAAM/XD2016 analyses. This study is partly supported by JST ERATO (JPMJER1301) and KAKENHI (17H01033, 17K05772 and 19H02552). We were granted access to the X-ray diffraction instruments in SPring-8 BL26B1 beamline (nos. 2018B2719 and 2019A1181) and KEK Photon Factory (2019G051).

## Author contributions

H.I. launched the project, K.I. and H.I. conceived the design, and K.I. and S.Y. synthesized the target. K.I., H.N., M.K. and H.I. developed defect descriptors. K.I., S.Y., S.S. and H.I. performed the crystallographic studies, and K.I. and S.Y. performed the DFT calculations. All authors analysed and discussed the results, and K.I. and H.I. wrote the manuscript.

## Competing interests

The authors declare no competing interests.
