## [Peer Review File · Nature Communications]

Reviewers' comments:

Reviewer #1 (Remarks to the Author):

This manuscript by Isobe et al. describes a nano-tube like molecule containing pyridine units as a nitrogen doped version of defect-involving nano-tube molecules reported previously by the authors group (Science 2019). The manuscript is well written and the supporting information is collected and arranged enough to support the results and discussion.

Unfortunately, the effects of nitrogen-doping on the structure and properties are rather small compared with the all carbon analog. I expected dramatic effects such as acid or metal responses, or reactivity. However, the structure is still very attractive as novel aromatic compounds, and this compound is scientifically valuable to report in Nature Communications as soon as possible.

Reviewer #2 (Remarks to the Author):

ISOBE – NCOMMS-19-38614

Isobe et al. report the synthesis of another spectacular nanotube molecule using their remarkable phenine concept, whereby organic structures can be expanded through the replacement of sp²-hybridized carbon atoms are replaced with 1,3,5-trisubstituted benzene (phenine) units. The novelty of the present work is that some of the phenine units have been changed to 2,4,6-trisubstituted pyridine units to afford an expanded Vögtle belt that contains 8 skeletal nitrogen atoms. As such, it is the first example of an N-doped nanotube segment. This is significant because it provides clear experimental information about the effects of N-doping in nanotubes, which has been lacking until now.

From a strategic perspective, the synthesis of the title compound proceeded essentially as it did for the previously reported hydrocarbon. I find this to be a point of genuine strength. It lends credence to the notion that the phenine approach to the construction of structurally elaborate and highly interesting molecules is not only robust, but also full of promise. All too often, the synthesis of a new and attention-grabbing aromatic molecule is a one-off, never to be repeated or built upon. What I see from the synthesis of the target molecule is a convincing demonstration of just how powerful this particular combination of Pt-based macrocyclization, Suzuki-Miyaura and Yamamoto couplings is. There is every reason to believe that bigger and even more amazing compounds are within reach. How about 1,3,5-triazine units?

I am impressed with how clean the spectra of all the synthetic intermediates are. This and the excellent synthetic procedures are hallmarks of first-rate synthetic chemistry. The title molecule has been obtained in pure form and has also characterized very well. The authors have gone to great lengths to obtain high quality X-ray structural information and achieving this is another laudable aspect of this work.

As a whole, the supporting information is superb.

The electronic properties of Isobe's N-doped nanotube molecule are indeed unexpected and interesting. The upshot is that the introduction of the N atoms leads to a substantial lowering in energy of the lowest-lying unoccupied molecular orbitals, but has a much less pronounced effect on the highest-lying occupied molecular orbitals. As a result, there is a reduction of the HOMO-LUMO gap. I find this to be quite remarkable because it contrasts what happens when acenes are N-doped (Bunz, *Acc. Chem. Res.* 2015, 48, 1676). For the acenes, N-doping lowers both the HOMO and the LUMO more or less equally. It may be worth including this point in the revised manuscript.

Overall, I think that this work definitely clears the bar for publication in *Nature Communications*. Apart from some minor polishing of the writing, I have no corrections to recommend.

Reviewer #3 (Remarks to the Author):

I believe the crystallography support the conclusion of the paper, however, great care must be taken in not over interpreting the charge density information. The structure contains disorder solvent which was removed by SQUEEZE this effect the strong low angle data, being an approximation does introduce errors, and it is this data that also includes a large amount of the information on bonding density. This is then coupled with the disorder in the structure further degrading the results.

The author repeatedly use the term anomalous ("deviating from what is standard, normal, or expected") to describe the disorder in this system. However, I would argue that is in not anomalous and is in fact the expected disorder in this system. If you consider the way neighboring tube interact, at the ends, through the interlocking of the methyl groups the observed rotation of the disorder component is such that its methyl group are superimpose on those of the major component, and so do not upset the packing and therefore is favorable. To me is more of a surprise that is not 50:50, but then this is just one crystal.

I would like the authors to include the CIF files for the SHELXL refinements with embedded res and HKL files to allow the reader the chance to really see how the multi-pole refinement compares. In the manuscript and SI only the R-values are given it would nice to see electron density map from SHELXL after SQUEEZE to see the residual electron density, 12% of a carbon atom should show up as around a 0.6 electron peak.

The comparison of bond lengths and average bond length are meaningless unless the su's are given, i.e C-N 1.34(2) and C-C 1.39(2) are statistically the same where as C-N 1.340(5) and C-C 1.390(5) are different.

Point-by-point response. Our responses are shown in blue.

Reviewer #1 (Remarks to the Author):

This manuscript by Isobe et al. describes a nano-tube like molecule containing pyridine units as a nitrogen doped version of defect-involving nano-tube molecules reported previously by the authors group (Science 2019). The manuscript is well written and the supporting information is collected and arranged enough to support the results and discussion.

We thank this reviewer for his/her precious time spared on evaluations. We also found his/her comment important to improve our work.

Unfortunately, the effects of nitrogen-doping on the structure and properties are rather small compared with the all carbon analog. I expected dramatic effects such as acid or metal responses, or reactivity.

We agree and now add new relevant data. While waiting for comments of reviewers/editors over 75 days, we were motivated to study the effects of nitrogen atoms and indeed examined acid-base reactions with our molecule. We first found that NpNT was fluorescent with a quantum yield of 16%. We then found that addition of trifluoroacetic acid to the specimen lowered the quantum yield to 8%. These new experimental data are now included in the main text (e.g., Fig. 4a and 4d) with additional experimental tales describing details.

A copy of Fig. 4:

However, the structure is still very attractive as novel aromatic compounds, and this compound is scientifically valuable to report in Nature Communications as soon as possible.

We thank this reviewer for his/her support for publication.

Reviewer #2 (Remarks to the Author):

ISOBE – NCOMMS-19-38614

Isobe et al. report the synthesis of another spectacular nanotube molecule using their remarkable phenine concept, whereby organic structures can be expanded through the replacement of sp^2 -hybridized carbon atoms are replaced with 1,3,5-trisubstituted benzene (phenine) units. The novelty of the present work is that some of the phenine units have been changed to 2,4,6-trisubstituted pyridine units to afford an expanded Vögtle belt that contains 8 skeletal nitrogen atoms.

As such, it is the first example of an N-doped nanotube segment. This is significant because it provides clear experimental information about the effects of N-doping in nanotubes, which has been lacking until now. From a strategic perspective, the synthesis of the title compound proceeded essentially as it did for the previously reported hydrocarbon. I find this to be a point of genuine strength. It lends credence to the notion that the phenine approach to the construction of structurally elaborate and highly interesting molecules is not only robust, but also full of promise. All too often, the synthesis of a new and attention-grabbing aromatic molecule is a one-off, never to be repeated or built upon. What I see from the synthesis of the target molecule is a convincing demonstration of just how powerful this particular combination of Pt-based macrocyclization, Suzuki-Miyaura and Yamamoto couplings is. There is every reason to believe that bigger and even more amazing compounds are within reach. How about 1,3,5-triazine units?

We thank this reviewer for his/her kind praise on our work. As indicated in the final sentence of our manuscript, many other heteroatom units are being included in our group. We hope that we can further entertain this reviewer, as well as readers, with these new molecules in the near future.

I am impressed with how clean the spectra of all the synthetic intermediates are. This and the excellent synthetic procedures are hallmarks of first-rate synthetic chemistry. The title molecule has been obtained in pure form and has also characterized very well. The authors have gone to great lengths to obtain high quality X-ray structural information and achieving this is another laudable aspect of this work.

As a whole, the supporting information is superb.

We are delighted to receive these kind comments on the quality of synthetic and crystallographic data. We hope that our works can positively contribute to the improvement of synthetic and structural chemistry.

The electronic properties of Isobe's N-doped nanotube molecule are indeed unexpected and interesting. The upshot is that the introduction of the N atoms leads to a substantial lowering in energy of the lowest-lying unoccupied molecular orbitals, but has a much less pronounced effect on the highest-lying occupied molecular orbitals. As a result, there is a reduction of the HOMO-LUMO gap. I find this to be quite remarkable because it contrasts what happens when acenes are N-doped (Bunz, Acc. Chem. Res. 2015, 48, 1676). For the acenes, N-doping lowers both the HOMO and the LUMO more or less equally. It may be worth including this point in the revised manuscript.

We thank this reviewer for turning our attention to this Accounts. We added a relevant sentence with this reference, which reads: "The effects of nitrogen atoms over the electronic structures of NpNT were unexpected from HOMO/LUMO-stabilizations found with N-doped acenes."

Overall, I think that this work definitely clears the bar for publication in Nature Communications. Apart from some minor polishing of the writing, I have no corrections to recommend.

We thank this reviewer for his/her strong support for publication.

Reviewer #3 (Remarks to the Author):

I believe the crystallography support the conclusion of the paper, however, great care must be taken in not over interpreting the charge density information.

We thank this reviewer for his/her time kindly spared for evaluations. We are particularly happy to learn that an expert of crystallography finds our conclusion valid.

Nonetheless, we fully agree with this reviewer in that charge density information should be carefully interpreted, which should be also emphasized for readers. As detailed below, we strengthened this aspect of our work by adding new data/information with due respect to valuable comments of this reviewer.

The structure contains disorder solvent which was removed by SQUEEZE. This effect on the strong low angle data, being an approximation, does introduce errors, and it is this data that also includes a large amount of the information on bonding density. This is then coupled with the disorder in the structure further degrading the results.

We fully agree with this summary including potential errors, which has urged us to provide detailed procedures in the original main text and Supplementary Information. With his/her valuable comments in mind, we wish to improve our work by including further details with new and supplemental data, as itemized below.

1. Supplemental cif file with res/HKL data

We now provide a raw cif file after the initial SHELXL refinement (named "stage 0") as CCDC 1984802 deposited at CCDC. With this file containing res and HKL data, the readers can examine and follow our procedures by themselves.

2. New additional data: deformation density map and electrostatic potential map for 4,4'-bipyridine

We now include deformation density map and electrostatic potential map for 4,4'-bipyridine in Supplementary Fig. 8. The data were similar to those of NpNT, which supported the validity of SQUEEZE/TAAM procedures newly devised for NpNT. Some relevant descriptions were also included in Supplementary Methods section.

A copy of **Supplementary Fig. 8:**

The author repeatedly use the term anomalous ("deviating from what is standard, normal, or expected") to describe the disorder in this system. However, I would argue that is in not anomalous and is in fact the expected disorder in this system. If you consider the way neighboring tube interact, at the ends, through the interlocking of the methyl groups the

observed rotation of the disorder component is such that its methyl group are superimpose on those of the major component, and so do not upset the packing and therefore is favorable. To me is more of a surprise that is not 50:50, but then this is just one crystal.

We agree in that we were a little too excited to find the rotational disorders. We also failed to clearly convey "our standard" for this excitement: as can be found in ref. 11, a previous carbon version, pNT, did not show this rotational disorder despite its close structural relevance to the present molecule. We thus added sentences to describe this fact and also removed most of "anomalous" from the text (four out of five).

I would like the authors to include the CIF files for the SHELXL refinements with embedded res and HKL files to allow the reader the chance to really see how the multi-pole refinement compares.

This is an excellent suggestion. We agree to provide res and HKL data. The cif file including these data after the initial SHELXL refinement (stage 0) is now provided as CCDC 1984802 so that the readers can examine and follow our SQUEEZE/TAAM procedures by themselves.

In the manuscript and SI only the R-values are given it would nice to see electron density map from SHELXL after SQUEEZE to see the residual electron density, 12% of a carbon atom should show up as around a 0.6 electron peak. We included the electron density map in Supplementary Fig. 4 (stage 0), which demonstrated that minor carbon atoms of the rotational disorders were not found at this stage under our normal protocols.

A copy of **Supplementary Fig. 4**:

The comparison of bond lengths and average bond length are meaningless unless the su's are given, i.e C-N 1.34(2) and C-C 1.39(2) are statistically the same where as C-N 1.340(5) and C-C 1.390(5) are different.

To avoid misconceptions, we revised Supplementary Fig. 7 to include raw bond-length data.

A copy of **Supplementary Fig. 7:**

REVIEWERS' COMMENTS:

Reviewer #1 (Remarks to the Author):

In the revised manuscript, the authors added absorption and fluorescence spectra measured in the presence of an acid in Figure 4. Although the effect is not so large, these data support the significance of the introduction of nitrogen atoms into the nanotube like molecule. The authors also revised the manuscript suitably to the other comments given by the other reviewers, especially the reanalysis of the X-ray disorder.

This reviewer recommends that this revised manuscript is ready for publication.

Reviewer #3 (Remarks to the Author):

The authors have addressed my comments to my satisfaction.